# Galectin-3 Mediated Endocytosis of the Orphan G-Protein-Coupled Receptor GPRC5A

**DOI:** 10.3390/cells14191571

**Published:** 2025-10-09

**Authors:** Abdeldjalil Boucheham, Jorge Mallor Franco, Séverine Bär, Ewan MacDonald, Solène Zuttion, Lana Blagec, Bruno Rinaldi, Johana Chicher, Laurianne Kuhn, Philippe Hammann, Christian Wunder, Ludger Johannes, Hocine Rechreche, Sylvie Friant

**Affiliations:** 1Molecular and Cellular Biology Laboratory—MCBL, Department of Molecular and Cellular Biology, Nature and Life Sciences Faculty, University of Jijel, Jijel 18000, Algeria; 2Université de Strasbourg, CNRS, Génétique Moléculaire, Génomique, Microbiologie (GMGM), UMR7156, 67000 Strasbourg, France; j.mallor.f@gmail.com (J.M.F.); sbar@unistra.fr (S.B.); solene.zuttion@etu.unistra.fr (S.Z.); lblagec@lbl.gov (L.B.); b.rinaldi@unistra.fr (B.R.); 3Biotechnology Research Center-CRBt, BP E73. UV N°03 Nouvelle Ville Ali Mendjelli, Constantine 25000, Algeria; 4INSERM, Université de Strasbourg, UMRS_1112, CRBS (Centre de Recherche en Biomédecine de Strasbourg), 67000 Strasbourg, France; 5Institut Curie, Université PSL, U1339 INSERM, UMR3666 CNRS, Chemical Biology of Cancer Unit, 75248 Paris, France; ewan.macdonald@crbm.cnrs.fr (E.M.); christian.wunder@curie.fr (C.W.); ludger.johannes@curie.fr (L.J.); 6Plateforme Protéomique Strasbourg-Esplanade, Institut de Biologie Moléculaire et Cellulaire (IBMC), CNRS UAR 1589, 67084 Strasbourg, France; j.chicher@ibmc-cnrs.unistra.fr (J.C.); lauriane.kuhn@iphc.cnrs.fr (L.K.); p.hammann@ibmc-cnrs.unistra.fr (P.H.); 7SAIRPICO Project Team, Inria Center at University of Rennes, U1339 INSERM, Institut Curie, UMR3666 CNRS, PSL Research University, 75248 Paris, France

**Keywords:** Galectin-3, GPRC5A, endocytosis, GL-Lect driven, colorectal cancer cells

## Abstract

Galectins, a family of glycan-binding proteins, play crucial roles in various cellular functions, acting at both intracellular and extracellular levels. Among them, Galectin-3 (Gal-3) stands out as a unique member, possessing an intrinsically unstructured N-terminal oligomerization domain and a canonical carbohydrate-recognition domain (CRD). Gal-3 binding to glycosylated plasma membrane cargo leads to its oligomerization and membrane bending, ultimately resulting in the formation of endocytic invaginations. An interactomic assay using proteomic analysis of endogenous Gal-3 immunoprecipitates identified the orphan G protein-coupled receptor GPRC5A as a novel binding partner of Gal-3. GPRC5A, also known as Retinoic Acid-Induced protein 3 (RAI3), is transcriptionally induced by retinoic acid. Our results further demonstrate that extracellular recombinant Gal-3 stimulates GPRC5A internalization. In SW480 colorectal cancer cells, glycosylated GPRC5A interacts with Gal-3. Interestingly, while GPRC5A expression was upregulated by the addition of all-trans retinoic acid (ATRA), its endogenous internalization in SW480 cells was specifically triggered by extracellular Gal-3, but not by ATRA. This study provides new insights into the endocytic mechanisms of GPRC5A, for which no specific ligand has been identified to date. Further research may uncover additional Gal-3-mediated functions in GPRC5A cellular signaling and contribute to the development of innovative therapeutic strategies.

## 1. Introduction

Galectins are glycan-binding proteins (lectins) that bear at least one carbohydrate-recognition domain (CRD), which binds to β-galactosides. Galectins can function intracellularly and can also be secreted to bind to cell surface glycoconjugated receptors. Galectins are involved in many cellular functions, such as cell–matrix adhesion, cell interactions, signaling and membrane trafficking [1]. Given their dual extracellular and intracellular functions, galectins are regarded as major players in normal physiology as well as in different disease conditions such as oncogenic processes. Structurally, there are three subfamilies of mammalian galectins: prototype, tandem repeat and chimera. Galectin-3 (Gal-3) encoded by the *LGALS3* gene is the only member of the chimera group. Gal-3 is a 26 kDa lectin with a disordered N-terminal domain involved in inter- and intra-molecular interactions and a single CRD [2]. Expression of the *LGALS3* gene is altered in many types of cancer, and Gal-3 is identified as an effective drug target for cancer diagnostics, as well as for inflammatory and fibrotic diseases [3,4,5]. Recent data show that increased blood concentrations of Gal-3 are observed in patients with severe COVID-19, highlighting its importance as a prognostic factor and as a promising treatment target [6,7]. Gal-3 multimerization via the N-terminal domain results in clustering, a process promoted by Gal-3 binding to glycan ligands through its CRD [1]. Moreover, Gal-3 binding to specific glycoproteins and glycolipids allows the rearrangement of their organization in the membrane [8,9]. Gal-3’s critical role in endocytosis is achieved via its recruitment to membranes by binding glycosylated cargos such as CD44, CD98, β1-integrin, or lactotransferrin; next clustering of these cargo proteins and glycosphingolipids generates membrane bending and the formation of endocytic invaginations [10,11,12,13]. This glycolipid and galectin-dependent endocytic mechanism has been termed GlycoLipid-Lectin (GL-Lect)-driven endocytosis [14].

G protein-coupled receptors (GPCRs) constitute a large family of conserved membrane receptors with seven transmembrane domains. They are activated by a diverse range of extracellular molecules to mediate signal transduction pathways. Furthermore, their involvement in many physiological and pathophysiological cellular processes establishes them as the most common drug targets [15]. Based on sequence homology, several receptor subfamilies have been identified. Among them, the GPCR family C group 5 (GPRC5) receptors are classified as a subfamily with four members, GPRC5A-D, for which no specific ligands are described. GPRC5A was identified as an all-*trans*-retinoic acid (ATRA) upregulated gene initially termed RAIG1 retinoic-acid inducible gene 1, and later also termed RAI3 [16,17]. A recent chemoproteomic profiling study shows that microbiota-derived aromatic monoamines can bind to GPRC5A and stimulate GPRC5A–β-arrestin recruitment [18]. Increased expression of GPRC5A is associated with colon, pancreas, and prostate cancers and could serve as a candidate biomarker for accurate diagnosis and prognosis [19,20].

Building on the established role of Gal-3 in mediating the endocytosis of membrane-associated proteins, we performed a Gal-3 interaction study using mass spectrometry to identify potential Gal-3 binding partners. The identified GPRC5A was internalized after the addition of Gal-3 and might follow a similar mechanism to those reported for other Gal-3 binding partners.

## 2. Materials and Methods

### 2.1. Cell Culture

The HepG2 (hepatocarcinoma) and HeLa (cervical adenocarcinoma) cell lines, along with the colorectal adenocarcinoma cell lines SW480, Caco-2, DLD1, HT29, and HCT116, were kindly provided by Dr. Isabelle Gross (CRBS, Strasbourg). Cells were cultured in DMEM supplemented with 10% fetal calf serum (FCS), 1% penicillin-streptomycin, and L-glutamine (Gibco). Cultures were maintained at 37 °C in a humidified incubator with 5% CO_2_ and were passaged twice a week. For immunofluorescence, cells were seeded one day before the experiments. Transient transfection was performed using Lipofectamine^®^ 2000 (ThermoFisher, Carlsbad, CA, USA) according to the manufacturer’s instructions.

### 2.2. Purification of Recombinant Galectin-3

Human recombinant wild-type Gal-3 with C-terminal 6xHis tags was prepared as previously described [13]. Briefly, Gal-3 expression was induced at 20 °C in Rosetta2-pLysS using three L of LB-media with 60 μM IPTG overnight, and purified by cobalt resin (Thermo Fisher Scientific, Rockford, lL, USA) affinity chromatography and gel filtration (Superdex75 10/300) in PBS at pH 7.3. Small aliquots for single use were snap-frozen and stored at −80 °C.

### 2.3. HepG2 Transient Transfection by GFP-Gal-3 Plasmid

For co-immunoprecipitation, cells were transfected with EGFP–Gal-3 plasmid [21]. pEGFP-hGal3 was a gift from Tamotsu Yoshimori (Addgene plasmid # 73080). HepG2 cells were grown in 75 cm^2^ flasks to ~70% confluence. Transfections were performed using Lipofectamine 2000 (Thermo Fisher Scientific) mixed with 20 µg plasmid DNA in Opti-MEM (Thermo Fisher Scientific); after 6 h, Opti-MEM was replaced with DMEM. At 48 h post-transfection, cells were lysed and total proteins extracted. For immunofluorescence, cells were seeded in 6-well plates on sterile coverslips (or Lab-Tek chamber slides) two days before the assay.

### 2.4. Cell Lysis and Protein Extraction

Confluent cells were washed twice with ice-cold 1× PBS and lysed on ice for 5 min in buffer containing 50 mM Tris-HCl (pH 8), 50 mM NaCl, 1% NP-40, and a cOmplete™ EDTA-free protease inhibitor cocktail (MilliporeSigma, Mannheim Germany). Cells were scraped and incubated for 5 min on ice. The resulting lysates were centrifuged at 14,000× *g* for 15 min at 4 °C. The collected supernatant (total protein extract) was quantified using the Bradford Protein Assay reagent (Bio-Rad, Hercules, CA, USA).

### 2.5. Co-Immunoprecipitation Experiments

Two immunoprecipitation methods were used to pull down proteins: the µMACS Protein G magnetic beads isolation kit (130-071-101, Miltenyi Biotec, Bergisch Gladbach, Germany), the µMACS Anti-GFP kit (130-091-288, Miltenyi Biotec), and Gamma-Bind Plus Sepharose beads (71705800 AK, Cytiva, Marlborough, MA, USA). The µMACS protocol was chosen for its magnetic separation efficiency, while the Gamma-Bind protocol served as an alternative to confirm protein interactions.

For the µMACS protocol, 1.2 mg of protein lysate was incubated with 50 µL of µMACS Protein G magnetic microbeads (Miltenyi Biotec) preincubated with or without anti-Gal-3 antibodies (mouse monoclonal, sc-32790, Santa Cruz Heidelberg, Germany) in a total volume of 500 µL. In parallel, 50 µL of lysate was kept as the input fraction. After 1 h incubation at 4 °C, samples were loaded onto a µMACS column within a µMACS Separator magnetic field. Beads and bound proteins were retained, while flow-through fractions represented non-retained proteins. After extensive washing (5 × 200 µL washing buffer: 50 mM Tris-HCl, 50 mM NaCl, 0.1% NP-40, supplemented with cOmplete™ EDTA-free protease inhibitors, MilliporeSigma), elution was performed using 70 µL of preheated elution buffer (50 °C) provided in the kit. Fractions were analyzed by SDS-PAGE. The same protocol was used for anti-GFP co-immunoprecipitation, with identical protein amounts incubated with µMACS GFP beads; µMACS Protein G microbeads were used as a negative control.

For Gamma-Bind assays, Gamma-Bind Plus Sepharose beads were equilibrated with lysis buffer (3 × 500 µL, centrifugation at 500× *g*, 4 °C, 2 min) and incubated with or without anti-Gal-3 antibodies (mouse monoclonal, Santa Cruz, sc-32790) at 4 °C for 2 h. After washing (3 × 500 µL lysis buffer), 1.5 mg of protein lysate in 1 mL total volume was added to the beads and incubated at 4 °C for 1 h with gentle agitation. Following additional washes (5 × 500 µL lysis buffer), elution was performed with 5× Laemmli buffer, followed by incubation at 37 °C for 5 min. After centrifugation, the recovered supernatants were analyzed by SDS-PAGE.

### 2.6. Immunoprecipitation Experiments for Interactomic Study

Immunoprecipitation was conducted using µMACS Protein G microbeads as described above, except that the total volume was 1 mL instead of 500 µL, and elution was performed in 100 µL. The eluted fractions (beads and negative control) were used for mass spectrometry analysis.

### 2.7. Tunicamycin Treatment

Tunicamycin treatment was performed according to Lakshminarayan et al. (2014) [10]. Cells were seeded in DMEM until ~80% confluence, then treated with 12 µg/mL tunicamycin for 48 h, with media renewed at 24 h. DMSO was used as a negative control. Thereafter, cells were lysed and proteins extracted as described above. Protein extracts were resolved by SDS-PAGE.

### 2.8. Western Blot Analyses

Proteins were resolved on 12% acrylamide/bis-acrylamide (29:1) gels. After electrophoresis, proteins were transferred to a Protran^®^ nitrocellulose membrane (Amersham, Freiburg-Germany). 2,2,2-Trichloroethanol (TCE) incorporated into SDS-PAGE gels enabled stain-free fluorescent detection of proteins [22]. Membranes were blocked with 4% non-fat milk in PBST for 1 h at room temperature and then incubated overnight at 4 °C with the following primary antibodies: anti-GPRC5A (1:1000, MilliporeSigma, HPA007928), anti-Gal-3 (1:2000, ab76245-Abcam, Cambridge, UK), anti-RIP (1:1000, AB_397832, BD Biosciences, Franklin Lakes, NJ, USA), and anti-GAPDH (1:10,000, Abcam rabbit, ab181602). After three washes (10 min each in PBST), membranes were incubated with HRP-conjugated secondary antibodies (1:10,000, Thermo Fisher) for 1 h at room temperature. Proteins were detected using a homemade ECL substrate, and images were acquired with a ChemiDoc™ Touch imaging system (Bio-Rad).

### 2.9. Mass Spectrometry Analysis

Protein precipitation was carried out using 0.1 M ammonium acetate in absolute methanol, after which the pellets were solubilized in 50 mM ammonium bicarbonate. Proteins underwent reduction and alkylation with 5 mM dithiothreitol and 10 mM iodoacetamide, respectively, before being digested overnight with sequencing-grade trypsin from porcine origin (1:25, *w*/*w*, Promega, Madison, WI, USA). The peptide mixtures obtained were dried under vacuum and reconstituted in water containing 0.1% (*v*/*v*) formic acid (solvent A). One-sixth of each digest was analyzed by nanoLC-MS/MS using an Easy-nanoLC-1000 system coupled to a Q-Exactive Plus mass spectrometer (ThermoFisher) operated in positive ion mode. For each run, 5 μL of sample were injected onto a C18 trap column (75 μm ID × 20 mm nanoViper, 3 µm Acclaim PepMap; ThermoFisher) connected to a C18 analytical column (75 μm ID × 25 cm nanoViper, 3 µm Acclaim PepMap; ThermoFisher). Separation was achieved with a 160 min gradient of acetonitrile containing 0.1% formic acid at a flow rate of 300 nL/min. Data-dependent acquisition (DDA) was performed on the Q-Exactive Plus using Xcalibur (version v4.2) software (ThermoFisher). Full MS scans were recorded at a resolution of 70,000 (at m/z 200) across a mass range of 350–1250 m/z, with a maximum injection time of 20 ms and an AGC target of 3 × 10^6^. Up to 10 of the most intense multiply charged ions (≥2) were selected for fragmentation, using a maximum injection time of 100 ms, an AGC target of 1 × 10^5^, and a resolution of 17,500. To prevent repeated selection of the same ions, a dynamic exclusion of 20 s was applied during the peak selection.

### 2.10. Database Search and Mass Spectrometry Data Post-Processing

MS data were searched against the UniProtKB database (release 2016_08, 149870 forward sequences) with Human taxonomy. Database interrogation was carried out using the Mascot search engine (version 2.3, Matrix Science, London, UK) with a decoy strategy. Search parameters were defined as follows: carbamidomethylation of cysteines was specified as a fixed modification, while protein N-terminal acetylation, phosphorylation on serine/threonine/tyrosine, and methionine oxidation were considered as variable modifications. Enzymatic specificity was set to trypsin, allowing up to three missed cleavage sites. Mass tolerances were adjusted to 10 ppm for precursor ions and 0.02 Da for fragment ions, with the instrument type defined as “ESI-Trap.”

Mascot result files (.dat) were subsequently processed in the Proline software (version v1.4) suite (http://proline.profiproteomics.fr/ (accessed on 9 March 2020)). Protein identifications were accepted if the Mascot pretty rank was equal to 1 and if a 1% false discovery rate (FDR) threshold was met at both the peptide spectrum match (PSM) and protein set levels (based on score). Protein quantification across samples was estimated using the total number of assigned MS/MS fragmentation spectra. The co-immunoprecipitation data were compared with the data collected from multiple experiments against the negative control to identify significant differences, as previously described [23].

Mass spectrometry data analysis and visualization were performed using R version 4.4.3 [24] within the RStudio integrated development environment (version 2024.12.1.563; [25]). The packages tidyverse [26], ggplot2 [27], readxl [28], openxlsx [29], and ggrepel [30] were used for data manipulation and plotting, as well as Excel import/export, and non-overlapping label annotation. Proteins were quantified based on the number of identified spectra per replicate and for each protein; the mean peptide count was calculated across biological replicates in both anti-Gal-3 IP experiment (*n* = 4) and negative control without antibodies (*n* = 4) conditions. To compute the log_2_ fold change (log_2_FC), a pseudo-count of 0.1 was added to all values prior to ratio calculation in order to avoid division by zero or log transformation of zero values. An unpaired two-tailed Student’s *t*-test was performed for each protein to assess statistical significance between control and test conditions. The resulting *p*-values were transformed to −log_10_(*p*-value) for visualization, and proteins were considered significantly enriched if they met both criteria: log_2_FC ≥ 1 and *p*-value < 0.05. Finally, volcano plots were generated to visualize fold change against statistical significance. Thresholds for significance (log_2_FC ≥ 1 and *p* < 0.05) were marked with dashed blue (vertical) and green (horizontal) lines, and significantly enriched proteins were highlighted in red (Figure 1A).

The mass spectrometry proteomics data have been deposited to the ProteomeXchange Consortium via the PRIDE partner repository [31], under the dataset identifier PXD048507 and 10.6019/PXD048507.

### 2.11. Gal-3 Dependent Endocytosis Assays

Non-confluent SW480 cells plated in 6-well plates containing coverslips were incubated in cold DMEM containing 0.2% BSA and anti-GPRC5A antibody (MilliporeSigma, HPA046526) added for 10 min. Unbound antibody was removed by washing with PBS++ (1X PBS, 0.5 mM CaCl_2_, 0.5 mM MgCl_2_), followed by a rinse with ice-cold DMEM containing 2% BSA. To initiate endocytosis, cells were treated with 1 µg/mL of purified recombinant Gal-3 in prewarmed DMEM supplemented with 0.2% BSA and incubated at 37 °C for the indicated duration. The medium was then replaced with ice cold PBS++, and cells were incubated on ice. To remove surface-bound antibodies or Gal-3, subsequent washes were performed three times with 0.5 M glycine (pH 2.2), followed by a single wash with 200 mM lactose. Cells were fixed with 4% paraformaldehyde for 15 min at room temperature and permeabilized with 0.2% Triton-X100 for 8 min. Following three washes with PBS, cells were blocked with 5% FCS in PBS for 45 min at room temperature. Cells were washed with PBS and 2% FCS-PBS containing DAPI and IgG-conjugated AlexaFluor 568 secondary antibody (ThermoFisher) diluted 1/500 applied. After two final washes with PBS, coverslips were mounted in elvanol and imaging was performed using a Zeiss Axio Observer D1 fluorescence microscope or Zeiss LSM800 confocal microscope (×40 objective, Plateforme d’Imagerie du CRBS PIC-STRA, Strasbourg, France). Images were analyzed using ImageJ (1.54g version) and Fiji (1.54p version). The methodology is adapted from previously published works [10,32].

## 3. Results

### 3.1. Identification of GPRC5A as a Gal-3 Binding Partner

To identify binding partners of Gal-3, we performed an interactomic analysis of endogenous Gal-3 using an approach previously applied to the VPS15 membrane trafficking and autophagy effector [33]. HepG2 cells were lysed and subjected to immunoprecipitation (IP) with anti-Gal-3 antibodies or without antibodies as a negative control (*n* = 4). After extensive washing, the bound proteins were identified using mass spectrometry. The interactomic data were validated by spectral counts and statistical analysis (Appendix A). A scatterplot of −log_10_(*p*-value) against log_2_(fold change) was generated (Figure 1A) to visualize differentially enriched proteins. This analysis revealed significant enrichment of Galectin-3 (Gal-3, also known as LEG3) in the anti-Gal-3 IP samples compared to negative controls. Notably, this analysis also showed a significant enrichment of previously identified Gal-3 interacting partners, including Galectin-3 Binding Protein (Gal-3BP) [34], CD59 [35], and BASI (Basigin/CD147) [36]. Gal-3BP, also known as LGALS3BP, is a tumor-associated antigen of approximately 90 kDa. It is an extracellularly secreted glycoprotein and a well-characterized ligand of Gal-3 [34]. CD59, a GPI-anchored protein also known as Protectin, is an immunoregulatory receptor that protects human cells from complement-mediated damage [37]. BASI (Basigin), also known as CD147 or extracellular matrix metalloproteinase inducer (EMMPRIN), is a transmembrane glycoprotein belonging to the immunoglobulin superfamily and is highly expressed in human tumors [38]. These previously reported interactors associate physically with Gal-3 in a carbohydrate-dependent manner and contribute to immune modulation (Gal-3BP) [39], complement resistance (CD59) [35], and matrix remodeling (BASI/CD147) [40]. For the other significantly enriched interacting proteins, no experimental validation has been reported to date.

Focusing on transmembrane proteins, since Gal-3 is known to specifically interact with membrane receptors [14], our interactomic data also identified the orphan G-protein-coupled receptor GPRC5A, known as RAI3. This receptor, whose expression is induced by retinoic acid, was significantly enriched in the Gal-3 IP samples. To confirm the interaction, we performed a co-IP assay (Appendix A), which revealed that Gal-3 (26 kDa) interacts with the 40 kDa forms of GPRC5A as well as with higher molecular weight forms. This is consistent with findings by Greenhough and colleagues [41], who reported that GPRC5A exhibits multiple electrophoretic migration bands.

Although the interaction was confirmed, the overall abundance of GPRC5A in the IP fractions was low. Next, we transfected Hepg2 cells with GFP-Gal-3 plasmid and performed anti-GFP co-IP experiments, to further analyze the interaction between Gal-3 and GPRC5A. The data reveal that GPRC5A was co-immunoprecipitated with GFP-Gal-3 and that higher molecular weight forms of GPRC5A were observed in the IP fraction (Figure 1B). Next, we extended our analysis to additional cell types to investigate the interaction under conditions showing higher levels of GPRC5A glycosylated forms.

### 3.2. GPRC5A Glycosylation in SW480 Colon Cancer Cells

The expression level of the *GPRC5A* gene varies across cancer types, displaying increased expression in colon, pancreas, and prostate cancers, and decreased expression in lung cancer [20]. Accordingly, we analyzed GPRC5A and Gal-3 protein levels compared to the RIP protein (loading control) and a TCE staining (loading control) in cell lines derived from colorectal (SW480, HCT116, HT-29, DLD-1, Caco-2), gastric (AGS) and cervical adenocarcinoma (HeLa) cancers, as well as HepG2 hepatocarcinoma cells (Appendix A). Compared with HepG2 cells, most of these cancer cell lines, except the gastric AGS cells, exhibited higher levels of GPRC5A, especially the higher molecular weight forms, previously reported [42].

From these cells, we selected the SW480 human colon adenocarcinoma cells for further analyses. The GPRC5A protein level was increased in SW480 cells compared to HepG2 cells, and higher molecular weight forms of GPRC5A were observed (Figure 2A). Since Gal-3 (like other lectins) binds specifically to certain glycan patterns [36,43], the glycosylation status of GPRC5A was analyzed by tunicamycin treatment. Tunicamycin is a drug that blocks N-linked glycosylation of newly synthesized proteins. SW480 cells were incubated for 48 h in presence of tunicamycin or with DMSO as a negative control, and the electrophoretic migration pattern of GPRC5A was analyzed (Figure 2B). Upon tunicamycin treatment, GPRC5A proteins lack some higher molecular weight forms, corresponding to N-linked glycosylated GPRC5A proteins.

The interaction between Gal-3 and GPRC5A was analyzed in SW480 cells. We observed the interaction of Gal-3 with GPRC5A, with a stronger signal for the high-molecular-weight forms compared to the 40 kDa forms (Figure 3A). This interaction was further confirmed by co-immunoprecipitation experiments performed on SW480 cells transfected with GFP-Gal-3 plasmid (Appendix A). These two coIP experiments reveal a strong binding to Gal-3 of the high-molecular-weight forms encompassing the N-glycosylated GPRC5A compared to the 40 kDa forms. These results indicate that in SW480 colorectal cancer cells, the carbohydrate-binding lectin Gal-3 binds to glycosylated GPRC5A.

### 3.3. Addition of Retinoic Acid Leads to Increased Levels of GPRC5A Proteins

GPRC5A was previously characterized as an all-trans-retinoic acid (ATRA)-upregulated gene [16,17]. The authors used a 10^−6^ M ATRA concentration for 24 h and tested different cell lines: human small cell lung carcinoma (NC1-N417), human embryonic kidney fibroblast (HEK293), human SY5Y neuroblastoma (SY5Y), and human astrocytoma (1321N1) [17]. As retinoic acid signaling pathways often exhibit non-linear, biphasic, or threshold responses rather than classic monotonic dose-dependence [44], we tested different ATRA concentrations, ranging from 10^−6^ M to 10^−8^ M. Indeed, low or moderate ATRA levels may most effectively activate gene expression, while higher concentrations can activate negative feedback, induce receptor desensitization, or trigger alternative signaling pathways. To assess the effect of ATRA on GPRC5A protein expression, we treated SW480 cells with ATRA (10^−6^ M, 10^−7^ M and 10^−8^ M) for 24 h and analyzed GPRC5A protein levels (Figure 3B). Treatment with ATRA increased GPRC5A expression. Interestingly, GPRC5A response to ATRA is effective at the different tested concentrations and may reflect the interplay of multiple retinoic acid-responsive pathways. Additionally, higher molecular weight forms encompassing N-glycosylated GPRC5A, which interact with Gal-3, were further upregulated in SW480 cells.

### 3.4. GPRC5A Endocytosis Is Mediated by Extracellular Gal-3 Addition

GPRC5A is an orphan GPCR with no identified ligand and our results show that it interacts with Gal-3 galectin. As Gal-3 was shown to mediate endocytic uptake of several glycosylated receptors, we tested whether GPRC5A was internalized upon Gal-3 addition. Purified recombinant Gal-3 was added to the cells and endocytosis assays were performed as previously described [10]. In SW480 colorectal cancer cells (Figure 3B), the expression level of GPRC5A is increased by addition of 10^−7^ M retinoic-acid for 24 h. Therefore, we added or not all-trans-retinoic acid (10^−7^ M ATRA) to SW480 cells for 24 h and incubated the cells with anti-GPRC5A antibodies; after extensive washing to eliminate the unbound antibodies, the cells were incubated without Gal-3 (no Gal-3) or in presence of Gal-3 (+Gal-3) for 25 min at 37 °C prior to fixation, permeabilization and observation of the localization of GPRC5A by addition of AlexaFluor-coupled-secondary antibodies (Appendix A). Interestingly, while the addition of ATRA did not change the localization of GPRC5A, Gal-3 still seemed to mediate its endocytic internalization as indicated by the presence of intracellular puncta observed after 25 min of incubation in the presence of Gal-3 (arrows in Appendix A, and higher magnification in Figure 2B). These results show that GPRC5A interacts with Gal-3 and give us a hint towards a possible endocytic internalization of GPRC5A mediated by extracellular Gal-3 addition.

Following this observation, the hypothesis of GPRC5A endocytosis mediated by Gal-3 was further investigated. After 30 min incubation with GRPC5A antibodies on ice, following extensive washing to eliminate the unbound antibodies, the SW480 colorectal cancer cells were incubated at 37 °C for 25 min in the presence or absence of Gal-3 at a final concentration of 1 µg/mL. Confocal microscopy showed that after Gal-3 treatment for 25 min, GPRC5A was internalized, forming distinct cytosolic puncta (Figure 4A). Internalization of GPRC5A was quantified using Fiji by measuring the fluorescence intensity per cell (Figure 4B) on confocal images. The data, displayed on a box plot, show a significant increase in intracellular fluorescence upon Gal-3 exposure. Interestingly, the fluorescence per cell varies a lot between cells, which may be due to variability in GPRC5A expression or a difference in endocytosis activity between cells. To complete the analyses, the number of cells presenting intracellular fluorescent puncta were also counted on pictures taken in fluorescence microscopy. The data indicate that more cells show internalized GPRC5A signal upon Gal-3 addition (Figure 4C). However, in those cells with internalized GPRC5A, no difference in the number of dots could be observed. These results indicate that GPRC5A is internalized by addition of extracellular Gal-3.

## 4. Discussion

The GPRC5 receptor family, including GPRC5A, features a short extracellular N-terminal domain and as opposed to other members of GPRC class C, agonists bind to the extracellular domain instead of the N terminal domain [45,46]. As orphan receptors, GPRC5 members lack identified ligands, but are upregulated by retinoic acid (RA), which has led to their classification as retinoic acid-inducible genes (RAIG) [16,17]. Herein, we report for the first time that Gal-3, a carbohydrate binding protein, and the orphan G protein-coupled receptor GPRC5A physically interact. This interaction was validated in HepG2 hepatocarcinoma and also observed in colorectal SW480 cancer cell lines. Notably, SW480 cells exhibited a higher expression level of GPRC5A compared to HepG2 cells. This is of interest as the GPRC5A expression level was shown to vary depending on the cancer type, and increased expression was reported in colorectal cancers as being associated with tumor progression [20,47].

Gal-3 plays a key role as a regulator of glycosylated proteins, either by their retention in lattices or by internalization [48]. Moreover, Gal-3 (like other lectins) binds specifically to certain glycan patterns [36,43]; so only those GPRC5A proteins with the right sugar modifications will be coimmunoprecipitated with Gal-3. Comparative glycomic studies between HepG2 and SW480 cells were not performed. However, comparative glycomic analyses on HepG2 (hepato carcinoma) and LO2 (normal liver) cells show that HepG2 shares 211 common N-glycan IDs with LO2 cells, and there are 140 unique cell-surface N-glycans for HepG2 cells [49]. The *N*-glycosylation profile of 25 colorectal cancer cells (CRC) among those SW480 cells, was determined and the data show that SW480 and others display the highest levels of *N*-glycans, while other CRC contained low levels (8.3–9.9%) [50]. These glycomic analyses reveal that different cell types and cell lines have specific glycosylation patterns. Comparative glycomic analyses would allow us to reveal the differences between HepG2 and SW480 cells.

Previous studies have identified various plasma membrane proteins, including CD44 and β1 integrin as Gal-3 binding partners, whose endocytosis is Gal-3 dependent [10]. Moreover, recent data show that acute inhibition of Gal-3 strongly decreases α5β1 integrin endocytosis. However, under prolonged Gal-3 inhibition, α5β1 integrin internalization is restored via clathrin-mediated endocytosis [51]. In this study, we observed that GPRC5A internalization was induced by extracellular Gal-3, this cargo being an orphan G-protein-coupled receptor whose expression but not its endocytosis is stimulated by treatment with retinoic acid.

Endocytosis pathways enable cells to communicate with their extracellular environment, receive signals and internalize plasma membrane receptors and nutrients. Various endocytic pathways have been described, including clathrin-dependent and clathrin–independent mechanisms, which can operate simultaneously and may provide cellular functional advantages [52]. One such process is GlycoLipid-Lectin (GL-Lect)-driven endocytosis, which mediates the internalization of glycosylated transmembrane proteins such as CD44, β1 integrin or CD98/SLC3A2 [10,12]. GPRC5A is likewise glycosylated and undergoes additional modifications such as palmitoylation [53]. Palmitoylation is a hallmark of GPCRs, with approximately 80% of known GPCRs containing potential cysteine residues for this modification. Experimental evidence confirms palmitoylation for several GPCRs. Palmitoylation affects diverse roles, including G-protein coupling, modulation of endocytosis, and regulation of receptor phosphorylation and desensitization [54]. In transmembrane proteins such as CD44, palmitoylation is essential for lipid raft association, receptor endocytosis, and turnover [55]. Exploring such PTMs could elucidate the mechanisms governing GPRC5A internalization and signaling.

Additionally, interactions between GPRC5B and sphingomyelin synthase 2 (SMS2) have linked sphingolipid metabolism to insulin resistance and obesity [56]. Whether GPRC5A or other GPRC5 subfamily members similarly interact with lipid synthesis pathways remains unknown. Cholesterol, sphingolipids, and glycolipids, vital components of lipid rafts, may mediate GPRC5A recruitment and endocytosis [57,58].

Interestingly, exosomes from HT29 colorectal cancer cells contain GPRC5A, Gal-3BP and Tetraspanin 1 (TSPAN1) [59]. In our interactomic data, we identified GPRC5A and Gal-3BP as Gal-3 interacting proteins. Gal-3 is also targeted to exosomes for extracellular release. Indeed, Gal-3 harbors a conserved tetrapeptide motif P(S/T) AP in its N-terminal domain that directly interacts with Tsg101, a component of the endosomal sorting complex required for transport (ESCRT); exosomal secretion of Gal-3 requires Tsg101 and Vps4a [60]. Gal-3 silencing reduces β1-integrin export in exosomes, which is associated with diminished metastatic potential in breast cancer cells [61]. Whether Gal-3 similarly regulates GPRC5A export remains to be determined.

Since GPCRs are among the most successful druggable targets, identifying the endocytosis pathway of the orphan receptor GPRC5A may improve therapeutic and/or prognostic strategies for recurrent and refractory diseases, particularly cancers.

## 5. Conclusions

Our findings reveal GPRC5A as a novel interacting partner of Gal-3. Moreover, we demonstrate that Gal-3 induces GPRC5A internalization. The consequences of this receptor trafficking for cellular functions remain unknown. Comparative analysis with normal cells could clarify whether Gal-3 mediated the internalization of GPRC5A is tumor-specific, and how it influences tumor cell growth, migration potential, or signaling. Finally, the differential expression of glycosylated and non-glycosylated forms of GPRC5A across tumor cell lines highlights the importance of investigating post-translational modifications more closely to unravel their roles in the Gal-3 interaction, internalization mechanisms, and the functional consequences of receptor trafficking. Such studies will improve the understanding of the role of the orphan receptor GPRC5A in both normal and cancer cells and may give some hints for potential future therapeutic approaches.

## Figures and Tables

**Figure 1 cells-14-01571-f001:**
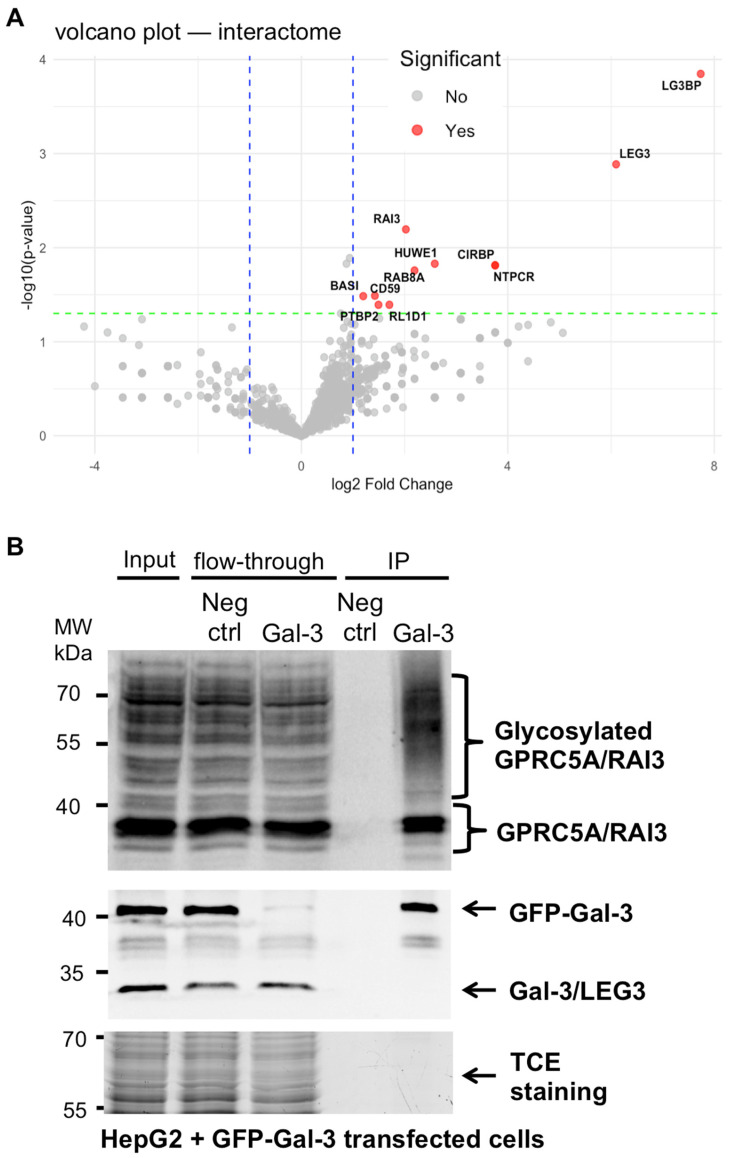
Identification of GPRC5A as a Gal-3 partner in HepG2 cells: (**A**) interactomic analysis of Gal-3 in HepG2 cells. Immunoprecipitation (IP) using anti-Gal-3 antibodies or a negative control (no antibody) was followed by nanoLC-MS/MS analyses. The results show the total number of identified spectra per protein across the different experiments. A volcano plot of −log_10_(*p*-value) versus log_2_(fold change) was generated to highlight proteins exhibiting significant differential detection between the IP (*n* = 4) and control (*n* = 4) conditions. Thresholds for significance (log_2_FC ≥ 1 and *p* < 0.05) are marked with dashed blue (vertical) and green (horizontal) lines, and significantly enriched proteins are highlighted in red. (**B**) Co-immunoprecipitation (co-IP) of GFP-Gal-3 (also termed LEG3) and GPRC5A (also termed RAI3). Total protein lysates (Input) were subjected to IP with anti-GFP antibodies using µMACS anti-GFP magnetic microbeads. The negative control using µMACS Protein G magnetic microbeads was performed in parallel. Immunoprecipitated (IP) and flow-through fractions were analyzed via Western blot with anti-GPRC5A (MilliporeSigma), anti-Gal-3 (AbCam) antibodies, and TCE staining was performed as loading control. The glycosylated forms of GPRC5A are indicated. TCE staining was used as a negative control. These data are representative of three independent experiments.

**Figure 2 cells-14-01571-f002:**
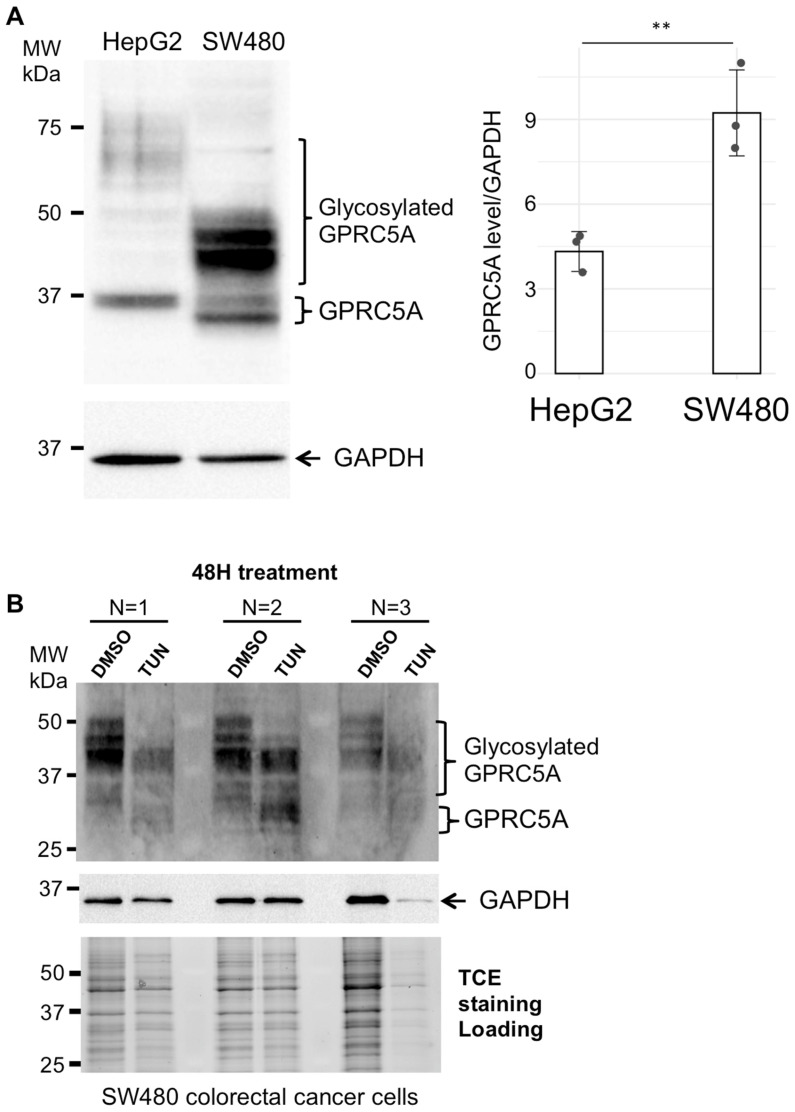
(**A**) GPRC5A expression level in HepG2 and SW480 cell lines. Western blot analysis using GPRC5A and GAPDH (loading control) antibodies were performed on total protein extracts from HepG2 and SW480 cells. These data are representative of three independent experiments. The GPRC5A protein level was quantified from three independent experiments. The mean GPRC5A level relative to GPADH was calculated and a *t*-test was performed. The *p*-value is the following, (**) *p* = 0.0086 < 0.01. (**B**) GPRC5A N-glycosylation in SW480 cells. SW480 cells were incubated for 48 h in presence of tunicamycin, an inhibitor of N-glycosylation or with DMSO as negative control. Next total protein lysates were subjected to Western blot analysis with anti-GPRC5A antibodies and anti-GAPDH and TCE staining as loading controls. The data from three independent experiments (N = 1 to N = 3) are shown.

**Figure 3 cells-14-01571-f003:**
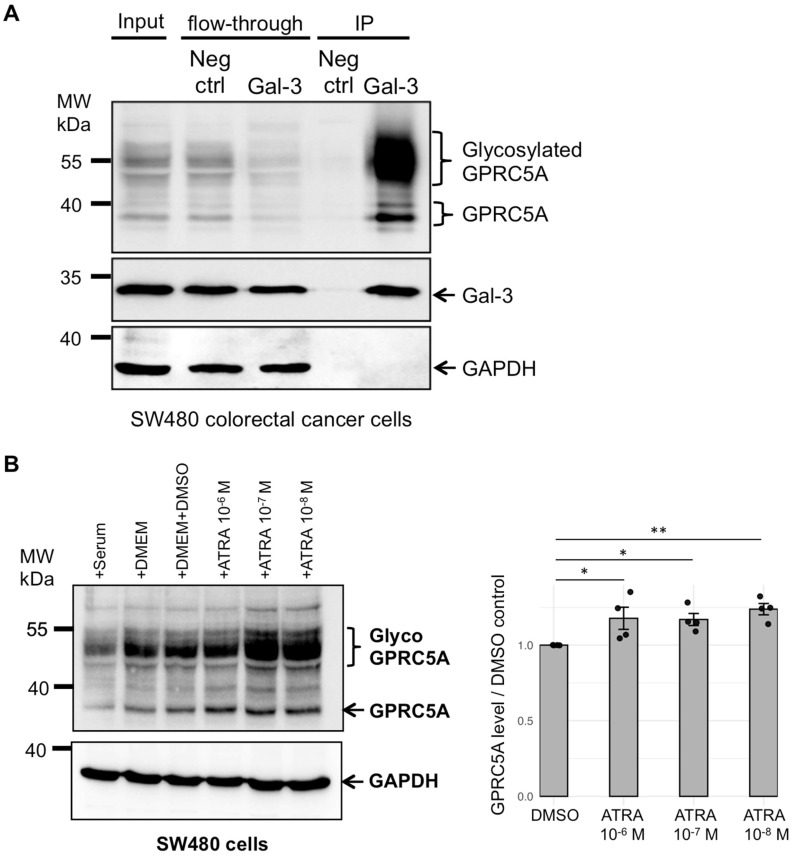
(**A**) Interaction between Gal-3 and GPRC5A in SW480 cells. Anti-Gal-3 immunoprecipitation (IP) or negative control IP without antibodies (Neg ctrl) using µMACS Protein G magnetic microbeads was performed on total protein extracts (Input) from SW480 cells. Western blot analysis was performed using antibodies against GPRC5A, Gal-3, or GAPDH (negative control). This experiment was performed one time. (**B**) GPRC5A protein level upon retinoic acid addition in SW480 cells. SW480 cancer cells were analyzed upon addition of the indicated components serum, DMEM, DMEM + DMSO (solvent for ATRA) or all-trans-retinoic acid (ATRA) for 24 h. Western blot analyses using anti-GPRC5A antibodies was performed on total protein extracts from treated cells. GAPDH was used as a negative control. GPRC5A glycosylated forms are indicated. These data are representative of four independent experiments. The level of GPRC5A protein was quantified upon ATRA treatment at the indicated concentration and compared to the control DMSO experiment, for the four independent experiments. The mean GPRC5A level (corrected with the loading control) relative to DMSO was calculated and a *t*-test was performed for each ATRA concentration. The *p*-values are the following: ATRA 10^−6^ M/DMSO (*) *p* = 0.046 < 0.05. ATRA 10^−7^ M/DMSO (*) *p* = 0.012 < 0.05. ATRA 10^−8^ M/DMSO (**) *p* = 0.0039 < 0.005.

**Figure 4 cells-14-01571-f004:**
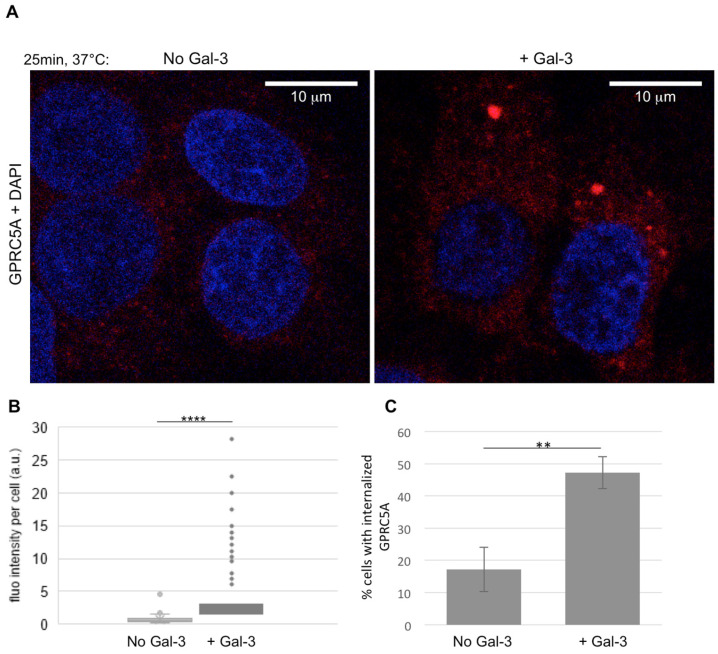
(**A**) SW480 colorectal cancer cells were incubated on ice for 30 min in the presence of anti-GPRC5A (Sigma, #:HPA046526) before transferring into a warm medium containing Galectin-3 (1 ug/mL) or not for 25 min (+Gal-3 or No Gal-3). After washing, fixation and incubation with 0.2% triton, the GPRC5A antibody was detected with an AlexaFluor568 coupled anti rabbit secondary antibody. Cells were stained with DAPI and pictures taken on a confocal (**B**) or fluorescence (**C**) microscope. (**B**) Red fluorescence intensity was measured using Fiji for 174 (no Gal-3) and 385 (+Gal-3) cells, the mean fluorescence was calculated and a Student test performed, (****): *p* = 2.2 × 10^−33^ < 0.00001. Data are represented as a box plot to show the variability between cells. (**C**) Cells were observed by fluorescence microscopy and the percentage of cells with red fluorescent dots within the cytoplasm was determined for three independent experiments. Mean percentage of cells with endocytosis of GPRC5A was calculated and a Student test performed (**) *p* = 0.0036 < 0.005.

## Data Availability

The mass spectrometric data were deposited on the PRIDE repository with the ProteomeXchange identifier PXD048507.

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
