# Peer review of "Galectin-3 Mediated Endocytosis of the Orphan G-Protein-Coupled Receptor GPRC5A"

_cells, 2025, doi:10.3390/cells14191571_

Round 1
Reviewer 1 Report
Comments and Suggestions for Authors
The authors described Gal-3 as a novel binding partner of the orphan GPCR GPRC5A. The scientific idea is rigorous an the experimental designs are thoughtful. Few minor changes needed before acceptance for publication.
- Line 129: 0,2 % Triton-X100 should be replaced with 0.2% Triton-X100
- Figure 2B; Do the author think majority of the GPRC5A does not interact with Gal-3 and comes out as flow through! please comment on that.
- There are no 'N' values for any of the figures except figure 1.
- Use of both protein G magnetic beads and Gamma-Bind Plus Sepharose beads definitely an wise choice for IP but it would have been more convenient if the author could IP down GPRC5A and looked for Gal-3 as interaction partner.
- Figure 3B: It is unclear why decreasing concentration of ATRA increases GPRC5A signal intensity.
- It would be interesting and to follow if PNGase F or tunicamycin treatment has any effect on the higher molecular weight GPRC5A signal bands
Author Response
The authors described Gal-3 as a novel binding partner of the orphan GPCR GPRC5A. The scientific idea is rigorous and the experimental designs are thoughtful. Few minor changes needed before acceptance for publication.
- Line 129: 0,2 % Triton-X100 should be replaced with 0.2% Triton-X100:
Done, corrected into the manuscript.
- Figure 2B; Do the author think majority of the GPRC5A does not interact with Gal-3 and comes out as flow through! please comment on that.
We thank the reviewer for this important point. Figure 1B shows that in protein extracts from HepG2 cells, most GPRC5A is in the flow-through, however this is not observed for SW480 cells (Figure 2B), with most GPRC5A being retained in the IP fraction. To explain this result, only the fraction that is properly glycosylated is competent to bind Galectin-3 (1). Moreover, Gal-3 (like other lectins) binds specifically to certain glycan patterns, so only those GPRC5A proteins with the right sugar modifications (2, 3) will be retained in the Gal-3-IP. Furthermore, GPRC5A–Gal-3 interactions that are weak, short-lived, or occur under specific conditions may not be co-precipitated and will remain in the flow-through. We have added this point into the Discussion of the revised manuscript.
References:
[1] Zhang M, Chen T, Lu X, Lan X, Chen Z, Lu S. G protein-coupled receptors (GPCRs): advances in structures, mechanisms, and drug discovery. Signal Transduct Target Ther. 2024;9(1):88. doi:10.1038/s41392-024-01803-6
[2] Joeh E, O'Leary T, Li W, et al. Mapping glycan-mediated galectin-3 interactions by live cell proximity labeling. Proc Natl Acad Sci U S A. 2020;117(44):27329-27338. doi:10.1073/pnas.2009206117
[3] Mattox DE, Bailey-Kellogg C. Comprehensive analysis of lectin-glycan interactions reveals determinants of lectin specificity. PLoS Comput Biol. 2021;17(10):e1009470. doi:10.1371/journal.pcbi.1009470
- There are no 'N' values for any of the figures except figure 1.
Done included into the manuscript
- Use of both protein G magnetic beads and Gamma-Bind Plus Sepharose beads definitely an wise choice for IP but it would have been more convenient if the author could IP down GPRC5A and looked for Gal-3 as interaction partner.
In our study, our primary aim was to identify Gal-3 interaction partners. For this reason, we chose to immunoprecipitate Gal-3 and then analyze associated proteins, among which GPRC5A emerged as a candidate. We thus considered GPRC5A as a downstream target, rather than the starting bait. We fully agree that reciprocal co-immunoprecipitation would provide valuable mechanistic insight into the nature of the Gal-3/GPRC5A interaction. However, our current focus was to establish that extracellular Gal-3 promotes GPRC5A internalization, which our results support. At this stage, the precise molecular mechanism and mode of interaction remain to be resolved, and we see reciprocal IP and mechanistic dissection as important directions for future work.
- Figure 3B: It is unclear why decreasing concentration of ATRA increases GPRC5A signal intensity.
We thank the reviewer for this valuable observation regarding the relationship between ATRA concentration and GPRC5A expression in Figure 3B. In our study, we compared two cells lines, and the maximal increase in GPRC5A levels was observed at 10-8M ATRA concentration. GPRC5A (RAIG3) was identified as a retinoic-acid induced gene, and the authors used a 10-6M ATRA concentration for 24 hr and tested the following cells lines: Cell lines studied were human small cell lung carcinoma (NC1-N417), human embryonic kidney fibroblast (HEK293), human SY5Y neuroblastoma (SY5Y), and human astrocytoma (1321N1) (1). As retinoic acid signaling pathways often exhibit non-linear, biphasic, or threshold responses rather than classic monotonic dose–dependence, we tested different ATRA concentrations, ranging from 10-6M to 10-8M. Indeed, low or moderate ATRA levels may most effectively activate gene expression, while higher concentrations can activate negative feedback, induce receptor desensitization, or trigger alternative signaling pathways. Further quantitative experiments (e.g., qRT-PCR, cell viability assays) could better dissect these mechanistic details and represent potential directions for future work. In the revised manuscript, we have added this point and we also clarified the ATRA concentrations used in the different assays.
Reference:
[1] Robbins MJ, Michalovich D, Hill J, et al. Molecular cloning and characterization of two novel retinoic acid-inducible orphan G-protein-coupled receptors (GPRC5B and GPRC5C). Genomics. 2000;67(1):8-18. doi:10.1006/geno.2000.6226.
- It would be interesting and to follow if PNGase F or tunicamycin treatment has any effect on the higher molecular weight GPRC5A signal bands
We thank you for this valuable suggestion. We agree that testing whether the higher-molecular-weight GPRC5A species correspond to N-glycosylated forms (e.g. by PNGase F treatment or tunicamycin) would provide useful mechanistic insight. In the revised manuscript, we have analyzed the effect of tunicamycin treatment and the data are included into the new Figure 2B.

Reviewer 2 Report
Comments and Suggestions for Authors
The manuscript is well-written, with a clear experimental flow and logical sequence of findings. However, I believe that several methodological details require clarification, and certain data presentation aspects should be improved to ensure reproducibility and enhance the manuscript's overall quality. Below are specific comments aimed at improving the manuscript.
Figure 1B
-The number of experiments is not mentioned. The authors need to show the raw data of at least 3 independent experiments. The same for Fig. 2B.
-How do the authors explain that, among all glycosylated forms of GPRC5A (shown in input and flow through), only a minor fraction of a rather less predominant form is co-immunoprecipated with Gal-3?
Figure 2B
The authors argue that, due to higher levels of expression of GPRC5A in SW480 cells, the levels of immunoprecipitation are better in these cells in comparison to HepG2 cells. Yet, the staining of GPRC5A in the input of SW480 cells in 2B appears to be much lower than the corresponding staining in HepG2 (in Fig. 1B). Despite the lower levels in the input of SW480, the levels of immunoprecipitated GPRC5A are higher in SW480 cells. How is this explained? The authors need to show in the same experiment, in the same blot, using equally loaded samples (mg of protein) the levels of these proteins in the input samples, and need to report the % of the IP sample used in the blot, in order to be able to directly compare the percentage of interaction between Gal3 and GPRC5A in these two cell types.
Fig. 3
It is not possible to assess how retinoic acid induces the expression of GPRC5A, as the time of incubation with retinoic acid is not mentioned. Is it due to reduction of degradation, or increased transcription?
Sup. Fig. 2
The puncta of increased signal in Gal3 treated cells could be either due to plasma membrane domains of GPRC5A multimers, or internalized vesicles. To discriminate between these two, the authors need to do IF experiments with non-permeabilized cells (the ab has no access to internal structures) versus permeabilized.
It is unclear whether the higher magnification images in Sup Fig 2B come from the field of view shown in A. The authors should indicate the corresponding areas in the low zoom images.
Quantification and statistics are required to prove that Gal-3 increases the levels of internalization.
The levels of GPRC5A do not appear to increase by the treatment with ATR (see Sup 2B). This contradicts the WB data (3B).
The confocal images presented raise several concerns regarding imaging conditions and data interpretation:
- Were all images acquired under identical confocal settings (laser power, gain, pinhole size, etc.)? This information should be provided to ensure data comparability.
- The quality of the images appear suboptimal. Was the fluorescence signal inherently weak (e.g., due to low antibody binding)? Methods such as line/frame averaging combined with adjusted laser intensity (within non-saturating levels) could improve image clarity.
- Details regarding the imaging acquisition should be included: Are the images single optical slices or projections of z-stacks? What was the z-step size?
Fig. 4.
The puncta could be unspecific staining. The authors need to prove the specificity of the ab, showing loss of the signal siRNA against GPRC5A treated cells.
The number of spots does not appear to increase in the presence of Gal-3, as in untreated cells the number of spots appears to be of the same levels. The intensity appears to be higher.
Please provide details on the quantification methodology: What regions of the cell were measured? What was the measurement area size? At which z-plane? Were acquisition parameters kept constant across conditions? Raw images should be provided.
In panel C, how many cells were analyzed and what was the criterion for defining a punctum as an intracellular red fluorescent spot? Given that the cytoplasmic staining appears diffuse with multiple small spots, how did the authors discriminate specific puncta for quantification (e.g., based on size threshold)? The authors should provide representative raw images used for this analysis and elaborate on the quantification methodology applied (e.g., automated spot detection, manual counting, threshold parameters).
Other points
Could the authors specify the total volume in which the 1.2 mg of lysate was diluted during the immunoprecipitation? The described washing steps (3x 500 μl) may not be sufficient to eliminate non-specifically bound proteins. This could explain the presence of numerous low log2FC values and high p-values observed in the Volcano Plot (Fig. 1A), which likely correspond to non-specific binders that were not efficiently removed. The authors should discuss this possibility and consider optimizing washing conditions.
Author Response
Reviewer 2
Figure 1B
-The number of experiments is not mentioned. The authors need to show the raw data of at least 3 independent experiments. The same for Fig. 2B.
All raw data were provided as a separate file at the time of submission of the manuscript, unfortunately this file was not made available to the reviewer. The number of experiments is included into the revised manuscript.
Figure 1B: How do the authors explain that, among all glycosylated forms of GPRC5A (shown in input and flow through), only a minor fraction of a rather less predominant form is co-immunoprecipated with Gal-3?
Figure 2B: The authors argue that, due to higher levels of expression of GPRC5A in SW480 cells, the levels of immunoprecipitation are better in these cells in comparison to HepG2 cells. Yet, the staining of GPRC5A in the input of SW480 cells in 2B appears to be much lower than the corresponding staining in HepG2 (in Fig. 1B). Despite the lower levels in the input of SW480, the levels of immunoprecipitated GPRC5A are higher in SW480 cells. How is this explained?
We thank the reviewer for this important point that was also raised by the other reviewer. Figure 1B shows that in protein extracts from HepG2 cells, most GPRC5A is in the flow-through, however this is not observed for SW480 cells (Figure 2B), with most GPRC5A being retained in the IP fraction. To explain this result, only the fraction that is properly glycosylated is competent to bind Galectin-3 (1). Moreover, Gal-3 (like other lectins) binds specifically to certain glycan patterns, so only those GPRC5A proteins with the right sugar modifications (2, 3) will be retained in the Gal-3-IP. Both cell lines likely have differences in glycosylation and post-translational modifications of GPRC5A, explaining this difference. The efficiency of Gal-3-mediated immunoprecipitation depends not only on total GPRC5A abundance, but mainly on the fraction of glycosylated GPRC5A susceptible to Gal-3 binding. Comparative glycomic studies between HepG2 and SW480 cells were not done. However, comparative glycomic analyses on HepG2 and LO2 cells show that HepG2 shares 211 common N-glycan IDs with LO2 cells, and there are 140 unique cell-surface N-glycans for HepG2 cells (4). The N-glycosylation profile of 25 colorectal cancer (CRC) cells (among those SW480 cells) was determined (5) and the data show that with 20.0% to 22.7%, SW480, SW620, SW48, Colo205_VUmc, and WiDr show that highest levels of N-glycans, while other CRC cell lines contained low levels (8.3–9.9%) (5). These glycomic analyses reveal that different cells types and cell lines have specific glycosylation patterns. Comparative glycomic analyses would allow to reveal the differences between HepG2 and SW480 cells. We have added this point into the Discussion of the revised manuscript.
Furthermore, GPRC5A–Gal-3 interactions that are weak, short-lived, or occur under specific conditions may not be co-precipitated and will remain in the flow-through.
References:
[1] Zhang M, Chen T, Lu X, Lan X, Chen Z, Lu S. G protein-coupled receptors (GPCRs): advances in structures, mechanisms, and drug discovery. Signal Transduct Target Ther. 2024;9(1):88. doi:10.1038/s41392-024-01803-6
[2] Joeh E, O'Leary T, Li W, et al. Mapping glycan-mediated galectin-3 interactions by live cell proximity labeling. Proc Natl Acad Sci U S A. 2020;117(44):27329-27338. doi:10.1073/pnas.2009206117
[3] Mattox DE, Bailey-Kellogg C. Comprehensive analysis of lectin-glycan interactions reveals determinants of lectin specificity. PLoS Comput Biol. 2021;17(10):e1009470. doi:10.1371/journal.pcbi.1009470
[4] Han Y, Xiao K, Tian Z. Comparative Glycomics Study of Cell-Surface N-Glycomes of HepG2 versus LO2 Cell Lines. J Proteome Res. 2019;18(1):372-379. doi:10.1021/acs.jproteome.8b00655
[5] Holst S, Deuss AJ, van Pelt GW, et al. N-glycosylation Profiling of Colorectal Cancer Cell Lines Reveals Association of Fucosylation with Differentiation and Caudal Type Homebox 1 (CDX1)/Villin mRNA Expression. Mol Cell Proteomics. 2016;15(1):124-140. doi:10.1074/mcp.M115.051235
- The authors need to show in the same experiment, in the same blot, using equally loaded samples (mg of protein) the levels of these proteins in the input samples, and need to report the % of the IP sample used in the blot, in order to be able to directly compare the percentage of interaction between Gal3 and GPRC5A in these two cell types.
We thank the reviewer and we corrected the manuscript by removing the sentences about the comparison of the percentage of interaction between Gal3 and GPRC5A in HepG2 and SW480 cells. Indeed, the data were obtained from different batches of cells, prepared at different times, and with different IPs and western blot. This shows that the GAL-3 and GPRC5A interaction can be observed in a different cell type than the one used for the interactomic analysis, but unfortunately does not allow direct comparison, what was not the aim of the study.
Fig. 3
It is not possible to assess how retinoic acid induces the expression of GPRC5A, as the time of incubation with retinoic acid is not mentioned. Is it due to reduction of degradation, or increased transcription?
Thank you for this important point. We apologize for omitting the incubation time from the Methods; we have now specified the ATRA treatment conditions in the Methods and figure legend.
GPRC5A (RAIG3) was identified as a retinoic-acid induced gene, and the authors used a 10-6M ATRA concentration for 24 hr and tested the following cells lines: Cell lines studied were human small cell lung carcinoma (NC1-N417), human embryonic kidney fibroblast (HEK293), human SY5Y neuroblastoma (SY5Y), and human astrocytoma (1321N1) (1).
In our study, we used the same concentrations and the same incubation times as described previously.
As retinoic acid signaling pathways often exhibit non-linear, biphasic, or threshold responses rather than classic monotonic dose–dependence, we also tested different ATRA concentrations, ranging from 10-6M to 10-8M. Indeed, low or moderate ATRA levels may most effectively activate gene expression, while higher concentrations can activate negative feedback, induce receptor desensitization, or trigger alternative signaling pathways.
Importantly, GPRC5A is a known retinoic-acid inducible gene (1), so the observed protein increase is consistent with transcriptional induction by ATRA; however, our current data set, based on protein detection only, cannot formally exclude post-transcriptional mechanisms such as altered protein stability or processing. We have clarified this limitation in the revised manuscript and propose RT-qPCR and protein stability assays as follow-up work to distinguish these possibilities.
Reference:
[1] Robbins MJ, Michalovich D, Hill J, et al. Molecular cloning and characterization of two novel retinoic acid-inducible orphan G-protein-coupled receptors (GPRC5B and GPRC5C). Genomics. 2000;67(1):8-18. doi:10.1006/geno.2000.6226.
Sup. Fig. 2
The puncta of increased signal in Gal3 treated cells could be either due to plasma membrane domains of GPRC5A multimers, or internalized vesicles. To discriminate between these two, the authors need to do IF experiments with non-permeabilized cells (the ab has no access to internal structures) versus permeabilized.
We apologize for the methods not being clear enough, indeed the permeabilization process was done only after the incubation with the primary anti-GPRC5A antibodies and after washing, to avoid staining of internal GPRC5A. The materials and methods section of the manuscript was corrected and clarified.
It is unclear whether the higher magnification images in Sup Fig 2B come from the field of view shown in A. The authors should indicate the corresponding areas in the low zoom images.
The images in B were not supposed to be a zoom of A since showing two different images seemed more accurate. However, the photo for the “+ Gal3, -ATRA” condition is a zoom, which was a mistake. Thus, the “+ GAL3; -ATRA” photo in B was replaced.
Quantification and statistics are required to prove that Gal-3 increases the levels of internalization.
The pictures in the supplementary figure are only to show that ATRA doesn’t seem to have an effect, the comment on GPCR5A internalization is only descriptive at this stage. Statistics were done on the pictures of the experiment in figure 4, confocal microscope ones for the fluorescence intensity and fluorescent microscope ones for the number of cells with fluorescent dots. In the revised manuscript, the paragraph was rephrased.
The levels of GPRC5A do not appear to increase by the treatment with ATR (see Sup 2B). This contradicts the WB data (3B).
The pictures in the supplementary figure are only to show that ATRA doesn’t seem to have an effect, the comment on GPCR5A internalization is only descriptive at this stage. Pictures presented in Sup. Fig 2 are not confocal images, they were taken on an epifluorescence microscope (keeping settings the same) which seemed more appropriate as a first approach, since we wanted to see all the dots present in the cell (not just one plane). The data show that without GAL-3 addition, there is an increase in GPRC5A signal after retinoic treatment, however after GAL-3 addition and internalization this increase in the GPRC5A signal is less clear. We did not follow up the analyses in presence of ATRA, as we did not observe obvious differences by microscopy observation. We thank the reviewer, as this could be an interesting follow up for future studies.
The confocal images presented raise several concerns regarding imaging conditions and data interpretation:
- Were all images acquired under identical confocal settings (laser power, gain, pinhole size, etc.)? This information should be provided to ensure data comparability.
- The quality of the images appear suboptimal. Was the fluorescence signal inherently weak (e.g., due to low antibody binding)? Methods such as line/frame averaging combined with adjusted laser intensity (within non-saturating levels) could improve image clarity.
- Details regarding the imaging acquisition should be included: Are the images single optical slices or projections of z-stacks? What was the z-step size?
Picture presented in supp fig 2 are not confocal, they are taken on a fluorescence microscope (keeping settings the same) which seemed more appropriate as a first approach since we wanted to see all the dots present in the cell (not just one plane). Going to the confocal microscope was planed as a second experiment to see the dots and their localization more in detail, but since the ATRA treatment didn’t have an effect, we didn’t go.
Statistics were done on the pictures of the experiment in Figure 4, confocal microscope ones for the fluorescence intensity and fluorescent microscope ones for the number of cells with fluorescent dots. Into the revised manuscript, the sentence was rephrased.
Fig. 4.
The puncta could be unspecific staining. The authors need to prove the specificity of the ab, showing loss of the signal siRNA against GPRC5A treated cells.
To determine whether the endocytosis of GPRC5A was specific, we transfected cells with a GPRC5A-Flag tag construct and analyzed the effect of GAL-3 addition. For endocytosis assays, transfection was performed using N-terminal Flag-GPRC5A plasmid, a modified GPRC5A-Tango construct containing a stop codon inserted by site-directed mutagenesis at the end of the GPRC5A coding sequence. GPRC5A-Tango was a gift from Bryan Roth (Addgene plasmid # 66382) (1). The data are shown in the response to reviewer file.
We also complemented the study, by a co-immunoprecipitation assay done on GFP-Gal-3 transfected cells, that is now shown in Fig1. B.
(1): Kroeze, W.K., M.F. Sassano, X.P. Huang, K. Lansu, J.D. McCorvy, P.M. Giguere, N. Sciaky, and B.L. Roth. 2015. PRESTO-Tango as an open-source resource for interrogation of the druggable human GPCRome. Nature structural & molecular biology. 22:362-369.
The number of spots does not appear to increase in the presence of Gal-3, as in untreated cells the number of spots appears to be of the same levels. The intensity appears to be higher.
Actually, yes, it’s the number of cells with spots which increases. In the cells with spots, no difference in the overall number of spots was seen but the Gal-3 treated cells have more brighter ones.
Please provide details on the quantification methodology: What regions of the cell were measured? What was the measurement area size? At which z-plane? Were acquisition parameters kept constant across conditions? Raw images should be provided.
Pictures are single slices, taken in the nucleus plane (focus done in the Dapi channel). Laser intensity and pinhole were kept the constant during the whole experiment and no adjustments made after pictures were taken.
The experiment was repeated to have a higher number of cells and 174 cells quantified for the “no Gal-3” and 385 for “+ Gal-3 “ condition (instead of 40 before).
For the quantification, cells were outlined by hand in Fidji (the area/ ROI size is thus variable since it depends on the cell size) and fluorescence measured for each cell. If cells were too close to each other to separate them properly, the ROI was drawn around the group of cells and fluorescence per cell calculated by dividing the fluorescence of the ROI by the number of cells (number of DAPI positive nuclei). To display the variability of fluorescence observed after addition of Gal-3 (which wasn’t that obvious before with probably too few cells measured but seemed interesting to point out), the histogram was replaced by a box plot.
The raw data including the excel file for quantification were provided as a separate file at the time of submission of the manuscript, unfortunately this file was not made available to the reviewer. All raw data are made available.
In panel C, how many cells were analyzed and what was the criterion for defining a punctum as an intracellular red fluorescent spot? Given that the cytoplasmic staining appears diffuse with multiple small spots, how did the authors discriminate specific puncta for quantification (e.g., based on size threshold)? The authors should provide representative raw images used for this analysis and elaborate on the quantification methodology applied (e.g., automated spot detection, manual counting, threshold parameters).
For counting the cells with dots, pictures used were mostly taken on a fluorescence microscope (mistake in the legend of Fig 4C - was corrected accordingly) to be independent of the Z. The only confocal picture used were the ones for which the fluorescence intensity was measured in fig 4B (one Z, at the level of the nucleus) which seemed interesting to compare intensity and number of cells at the time.
Since the endocytosis experiment was done again to take more confocal pictures anyway and to homogenize the experiments, pictures of the newly done experiment were also taken on a fluorescence microscope and positive cells counted (no Gal-3: 328 cells, Gal-3: 465 cells). The data obtained replaces the one from the confocal microscopy experiment used before. The graph has been replaced but the result is quite the same (17,2% pos cells for nogal, 47,2% for gal compared to 15,5% for nogal and 47,8% for gal before), the difference between the “no Gal-3” and “+ Gal-3” conditions still being significant.
The Puncta were those dots which presented a higher fluorescence intensity than the rest of the cytoplasm by eye (see example below, arrows point at the positive cells). The number of cells counted were: experiment 1: no Gal-3 328, Gal-3 465, experiment 2: no Gal-3: 93, Gal-3: 106, experiment 3: no Gal-3: 107, Gal-3: 228.
Other points
Could the authors specify the total volume in which the 1.2 mg of lysate was diluted during the immunoprecipitation? The described washing steps (3x 500 μl) may not be sufficient to eliminate non-specifically bound proteins. This could explain the presence of numerous low log2FC values and high p-values observed in the Volcano Plot (Fig. 1A), which likely correspond to non-specific binders that were not efficiently removed. The authors should discuss this possibility and consider optimizing washing conditions.
This is now corrected and included into the manuscript.
For the washing step for the interactomic study, we used the µMACS column within a µMACS Separator magnetic field and extensive washing (5 × 200 µl) with the following washing buffer: 50 mM Tris-HCl, 50 mM NaCl, 0.1% NP-40, supplemented with cOmplete™ EDTA-free protease inhibitors, MilliporeSigma).
This washing buffer contains 0.1% NP-40 detergent to efficiently remove the non-specific bound proteins. The aim of the analysis was to be stringent enough, to characterize specific Gal-3 interacting proteins; as in the statistical validated partners, we found some that were already known to interact with Gal-3, we considered the results to be accurate and further studied GPRC5A.

Round 2
Reviewer 2 Report
Comments and Suggestions for Authors
The manuscript is significantly improved, although not all my concerns have been addressed.